# Trends and projections of under-5 mortality in Bangladesh including the effects of maternal high-risk fertility behaviours and use of healthcare services

**Mostaured Ali Khan[1,2], Nuruzzaman Khan[3,4], Obaidur Rahman[5,6], Golam Mustagir[1], Kamal Hossain[1], Rafiqul Islam[1]\*, Hafiz T. A. Khan[7]**

1 Department of Population Science and Human Resource Development, University of Rajshahi, Rajshahi, Bangladesh, 2 MEL and Research, Practical Action, Dhanmondi, Dhaka, Bangladesh, 3 School of Public Health and Medicine, Faculty of Health and Medicine, The University of Newcastle, Callaghan, NSW, Australia, 4 Department of Population Sciences, Jatiya Kabi Kazi Nazrul Islam University, Mymensingh, Bangladesh, 5 Department of Global Health Nursing, Graduate School of Nursing Science, St. Luke's International University, Tokyo, Japan, 6 Department of Global Health Policy, School of International Health, The University of Tokyo, Tokyo, Japan, 7 Public Health & Statistics, College of Nursing, Midwifery and Healthcare, University of West London, Brentford, United Kingdom

\* rafique_pops@yahoo.com

## Abstract

### Objective

This study examines trends and puts forward projections of under-5 mortality (U5M) in Bangladesh and identifies the effects of maternal high-risk fertility behaviours and use of healthcare services.

### Methods

Data from seven waves of the Bangladesh Demographic and Health Survey (1994–2014) were analyzed for trends and projections of U5M and a Chi-square ($\chi^2$) test was used to identify if there was any association with maternal high-risk fertility behaviours and use of healthcare services. A multivariate logistic regression model was used to determine the effects of fertility behaviors and healthcare usage on the occurrence of U5M adjusting with confounders.

### Results

U5M declined from 82.5 to 41.0 per 1000 livebirths during 1994–2014 and is projected to further reduce to 17.6 per 1000 livebirths by 2030. The study identified a noticeable regional variation in U5M with maternal high-risk fertility behaviours including age at birth <18 years (aOR: 1.84, 95% CI: 1.23–2.76) and birth interval <24 months (aOR: 1.56, 95% CI: 1.02–2.37) found to be significant determinants. There was a 39–53% decline in this rate of mortality among women that had used antenatal care services at least four times (aOR, 0.51, 95% CI: 0.27–0.97), delivery care (aOR, 0.47, 95% CI: 0.24–0.95), and had received post-natal care (aOR, 0.61, 95% CI: 0.41–0.91) in their last birth. Cesarean section was found to

**Data Availability Statement:** The data underlying this study (BDHSs, 1993-2014) are owned by a third party. However, the authors had no special

access privileges and other researchers may access the data in the same manner from the MEASURE DHS Archive via the instructions included in the following link: http://dhsprogram.com/data/Using-Datasets-for-Analysis.cfm.

**Funding:** The author(s) received no specific funding for this work.

**Competing interests:** The authors have declared that no competing interests exist.

**Abbreviations:** U5M, Under-5 Mortality; NM, Neonatal Mortality; IM, Infant Mortality; LMIC, Low- and Middle-Income Country; BDHS, Bangladesh Demographic and Health Survey; ARR, Annual Rate of Reduction; ANC, Antenatal Care; SBA, Skilled Birth Attendant; LBW, Low Birth Weight; CI, Confidence Interval; OR, Odds Ratio.

be associated with a 51% reduction in U5M (aOR, 0.49, 95% CI: 0.29–0.82) compared to its non-use.

## Conclusion

The Sustainable Development Goals require a U5M rate of 25 per 1000 livebirths to be achieved by 2030. This study suggests that with the current trend of reduction, Bangladesh will achieve this target before the deadline. This study also found that maternal high-risk fertility behaviours and non-use of maternal healthcare services are very prevalent in some regions of Bangladesh and have increased the occurrence of U5M in those areas. This suggests therefore, that policies and programmes designed to reduce the pregnancy rates of women that are at risk and to encourage an increase in the use of maternal healthcare services are needed.

## Introduction

The Sustainable Development Goals (SDG) are focused (Target 3.2) on reducing neonatal mortality (NM) and under-five mortality (U5M) rates to 12 and 25 deaths per 1,000 live births respectively by 2030 [1]. Success in meeting these aims will require significant reductions in current rates in low- and lower-middle-income countries (LMICs), particularly in Sub-Saharan Africa (77.5 and 27.7 deaths per 1000 live births for U5M and NM, respectively) and South Asian countries (42.1 and 25.8 deaths per 1000 live births for U5M and NM, respectively) where the majority of current child deaths occur [2]. The Millennium Development Goals (MDG) made little difference towards such a reduction in LMICs between 2000 and 2015 [3, 4]. For instance, only 13 of the 95 LMICs achieved the MDG's target reduction in the U5M rate, though noticeable progress was reported in other countries (at around 53%) [3–6]. Bangladesh, a lower-middle-income country of South Asia, also achieved a significant reduction in U5M (from 133 to 30.2 deaths per 1000 live births between 1990 and 2018) following the successful implementation of the MDG [3, 7]. However, the rates are still high among South Asian countries and Bangladesh is in 3rd position after Pakistan (69.3 per 1000 live births) and India (36.6 per 1000 live births) [8, 9]. There is no doubt that extensive work and continued efforts are important for ensuring further reductions of NM and U5M in order to achieve the respective SDG targets.

There are numerous reasons for the higher rates of NM and U5M in LMICs, however, rurality is the important one [1, 10, 11]. The causes are higher illiteracy and lower availability and utilisation of maternal healthcare services including antenatal care (ANC), delivery care, and postnatal care [12]. Moreover, early marriage and adolescent fertility are higher in LMICs, particularly in rural areas, that serve to increase the occurrence of U5M. These groups of women are very likely to experience nutritional disorders including undernutrition and anemia, pregnancy complications, and birth defects [5, 10, 13–15]. These disorders are independent risk factors that can lead to adverse birth outcomes, along with preterm birth and low-birth weight, as well as NM and U5M [16, 17]. Socio-demographic risk factors, such as the low education of a woman's partner, plus women engaged in marginal or lower-paid jobs, and poor household wealth are also frequently mentioned as determinants of U5M in LMICs [10, 14, 15]. Certain birth-characteristics have been identified by a number of studies around the death of children such as U5M being higher among girls, among children with a history of

sibling deaths, and among children born within <2 years of their most-recent sibling [10, 18, 19]. The non-use of tetanus toxoids vaccination is another important factor for U5M and NM [20, 21]. The World Health Organization (WHO) recommends that a 5% usage of cesarean sections (CS) could ensure better maternal and child health outcomes [22], but unnecessary overuse of CS (higher than 15%) could be responsible for higher rates of still births and perinatal deaths, with this particular burden significantly higher in LMICs [23]. These figures indicate that reaching a further reduction in the U5M rate is challenging and needs consideration of myriad factors and preferably a multi-sectoral approach.

Bangladesh is currently implementing a range of programmes that include improving accessibility of maternal healthcare services and vaccine coverage in order to help achieve the SDG targets for NM and U5M. However, there has been no systematic assessment of the success or otherwise of these programmes and also a lack of studies that project the future occurrence of NM and U5M that is important for policymakers in devising evidence-based policies. Of the few studies that looked into child-mortality, a number of specific risk factors were identified such as birth order, birth interval, child's gender, maternal age at birth, parental education and employment status, delivery complications, and anemia [7, 13, 18, 20]. However, none of these studies considered spatial variations of U5M that is higher in Bangladesh and could be key for devising area-targeted policies to enhance their effectiveness [24]. Maternal high-risk fertility behaviours (age at birth >18 year, birth interval >24 months, birth order >3) [17, 25] and lower usage of maternal healthcare services are common in Bangladesh [7, 17, 18, 26] and could be major contributors to the occurrence of U5M. Studies conducted in LMICs repeatedly reported these effects [1, 13, 19, 27–29] but there is no clear evidence available in the case of Bangladesh. This study addresses this gap and its aims are: (i) to project the reduction rate of U5M up to 2030 based on current estimates, (ii) to show spatial variations in the occurrence of U5M in Bangladesh, and (iii) to assess the effects of maternal high-risk fertility behaviours and usage of maternal healthcare services on U5M.

## Methods

### Study design

Data for this study were extracted from seven waves of the Bangladesh Demographic and Health Surveys (BDHSs) conducted between 1994 and 2014. These cross-sectional surveys were carried out every three years in each of the seven administrative divisions of Bangladesh [30]. In each round, all married women aged 15–44 years in a nationally representative sample of households were interviewed using a two-stage stratified cluster sampling design. In the first stage, a fixed number of primary sampling units (PSU) were selected with the probability proportional to the unit size. In the second stage, a total of 30 households was selected in each primary sampling unit through systematic random sampling [30]. Structured questionnaires were used to collect data on respondents' background, community-level factors, and different health issues including maternal health, child health, family planning, and use of maternal healthcare services. The National Institute of Population Research and Training (NIPORT) conducted all these surveys and there was a 98% overall response rate for each survey. Broad descriptions regarding each of the BDHSs is published elsewhere [30–36]. The BDHS data is publicly available upon request on: https://dhsprogram.com/

### Analysis sample

The data from a total of 46,592 children aged below five years was analysed for this study covering each of the following years: 1994 (n = 3,874); 1997 (n = 6,189); 2000 (n = 6,832); 2004 (n = 6,908); 2007 (n = 6,150); 2011 (n = 8,753); 2014 (n = 7,886). This sample was generated

from interviews with 87,452 women in seven different surveys conducted in each of the following years: 1994 (9,640); 1997 (9,127); 2000 (10,544); 2004 (11,440); 2007 (10,996); 2011 (17,842); and 2014 (17,863) [30–36]. The criteria for collecting this data involved women that had complete birth histories that had occurred five years prior to the survey date (whether each child was alive at the time of the survey or not), including the month and year of the child's birth, and how old a child was if s/he had died. These retrospective data sets were used to identify the number of children born in the last five years and age of children at death.

## Outcome variable

Neonatal Mortality (NM), infant mortality (IM) and Under 5 Mortality (U5M) are considered as the outcome variables for this study. The BDHSs recorded this information by asking women "*Did you give birth to any children within five years*?" and if the answer was yes, they were then asked "*Is the child alive*?". Respondents were asked similar follow-up questions for every child if they had given birth to multiple children within this time frame. If death was recoded to this item (for every child; single or multiple) then the follow-up question was asked in order to record timing of death (in days). For this study, these responses were placed into three categories: NM (if death occurs within first 28 days of birth), IM (if death occurs before reaching the first birthday), and U5M (if death occurs before reaching the fifth birthday).

## Predictor variables

Maternal high-risk fertility behaviours and use of healthcare services were considered as main predictor variables for under-five mortality. For this situation, a dichotomous variable on maternal high-risk fertility behaviours (1, yes and 0, no) was generated where positive responses were considered if women reported they were aged <18 years in their most recent birth, or had a birth interval <24 months in their most recent successive births, or were aged >34 years at their last birth, or had birth order >3. Another generated dichotomous variable covered multiple high-risk fertility behaviours (1, yes and 0, no) where a positive was coded if women reported at least two of the previously mentioned characteristics. The WHO's guidelines were used to classify use of healthcare services covering antenatal (<4, ≥4), delivery (yes, no), and postnatal care (yes, no) services [14, 18, 25, 30, 37, 38]. The WHO changed their guideline on ANC in 2016 with a new recommendation of eight visits but for this study, the pre-2016 guideline was followed as data up to the year 2014 was used. A complete list of explanatory variables is presented in Table 1.

## Confounding adjustment

In this study, the effects of maternal high-risk fertility behaviours and use of healthcare services on the prevalence of U5M were determined through adjusting different individual, household/ family and community level characteristics that were selected by reviewing the literature [14, 18, 25, 30, 37, 38].

## Statistical analysis

A linear regression model was used to project the U5M rate up to the year 2030, the year where the current round of development goals, that is, the SDG are due to be achieved. The model is $y = a_0 + a_1 x + e$; where $y$ = predicted rate, $x$ = given year, $a_0$ = the constant, $a_1$ = the regression coefficient and e = the standard error. The parameters of the model were identified through a time series ARIMA analysis. A robust nonlinear smoothing technique (4253H, twice) enabled adjustment for various unexpected distortions before completion of the

**Table 1. A complete list of explanatory variables.**

| Explanatory variables | Collected data | Answer category |
|---|---|---|
| **Socio-demographic variables** | | |
| Maternal current age (in year) | Maternal age at the time of the survey | 1 = 15–24 years; 2 = 25–34 years; 3 = 35–49 years |
| Maternal education[1] | Maternal highest level of education | 1 = No education; 2 = Primary; 3 = Secondary & above |
| Residence | Place of residence | 1 = Urban; 2 = Rural |
| Economic status[2] | Wealth index of the family | 1 = Poor; 2 = Middle; 3 = Rich |
| Employment status | Respondents are employed or not | 0 = Unemployed; 1 = Employed |
| **Unwanted birth** | The birth was not wanted at that time | 0 = No; 1 = Yes |
| **Very low birth weight (VLBW) of child** | The baby was born with very low birth weight (<1500gm) (according to mother's perception about size of the baby at birth was obtained. Very small baby size at birth considered as a useful proxy of LBW)[2] | 0 = No; 1 = Yes |
| **Maternal high-risk fertility behaviors** | | |
| Maternal age at birth <18 years | The mother whose age at the time of the birth was less than 18 years | 0 = No; 1 = Yes |
| Maternal age at birth >34 years | The mother whose age at the time of the birth was greater than 34 years | 0 = No; 1 = Yes |
| Birth interval <24 months | The mother who gave birth with a birth interval of less than 24 months | 0 = No; 1 = Yes |
| Birth order >3 | The mother whose birth order was higher than 3 | 0 = No; 1 = Yes |
| Maternal age at birth <18 years and Birth interval <24 months[3] | The mother whose age at the time of the birth was less than 18 years with an interval of less than 24 months | 0 = No; 1 = Yes |
| Maternal age at birth >34 years and Birth interval <24 months[4] | The mother whose age at the time of the birth was greater than 34 years with an interval of less than 24 months | 0 = No; 1 = Yes |
| Birth interval <24 months and birth order >3 | The mother whose birth order was higher than 3 with interval of less than 24 months | 0 = No; 1 = Yes |
| **Maternal healthcare utilization** | | |
| Taken ANC at least 4 times | The mother who had taken ANC at least 4 times during pregnancy | 0 = No; 1 = Yes |
| Institutional delivery | The delivery was taken place in a hospital or clinic etc. | 0 = No; 1 = Yes |
| Skilled birth attendants (SBA) | The baby was delivered by an SBA like doctor, nurse, trained health personnel etc. | 0 = No; 1 = Yes |
| Cesarean section (CS) | The delivery was a cesarean section delivery | 0 = No; 1 = Yes |
| Postnatal care (PNC) | The mother and baby had taken postnatal care (2 months) | 0 = No; 1 = Yes |

**Note:** The analysis was restricted for respondents (mothers) who have given birth within 5 years prior to the survey.

[1] Primary and secondary education is defined as completing grade 5 and 10, respectively.

[2] Followed standard BDHS measure.

[3] Includes the category age at birth <18 years with birth order >3 and age at birth <18 years with interval <24 months and birth order >3.

[4] includes the category age at birth <34 years with interval <24 months and age at birth <34 years with interval <24 months and birth order >3.

projection for U5M [39]. The overall decline in the U5M rate was then calculated by using the formula, $\frac{U5M_t - U5M_o}{U5M_0} \times 100$. The Annual Rate of Reduction (ARR) of U5M was calculated using the formula $r = \left( \sqrt[n]{\frac{U5M_t}{U5M_0}} - 1 \right) \times 100$; where $U5M_0$ = U5M rate of any given year; $U5M_t$ =

U5M rate of $t^{th}$ year; r = ARR; n = number of years between two surveys [40]. Chi-square tests (set at $p<0.05$ level of significance) helped determine if there were any associations between under-five mortality with the characteristics of maternal high-risk fertility behaviours and use of healthcare services. Finally, the effects of maternal high-risk fertility behaviours and use of healthcare services were determined through the multivariate logistic regression model. Bangladesh has seen rapid socio-economic developments over the last decade and so the multivariate analysis was restricted to the most recent BDHS 2014 data. The model was adjusted for regions, residence, education, economic status, unwanted birth, and very low birth weight (VLBW). Results were reported by odds ratios (ORs) with its 95% Confidence Interval (95% CI) and all analyses, taking into account complex survey design and sample weights of the surveys, were conducted using the statistical programme called Stata version 13.0 (Stata Corporation, College Station, TX, USA).

## Ethics statement

The study analysed secondary data of BDHS. The BDHS is a part of the worldwide Demographic and Health Surveys (DHS) programme. The Bangladesh Ministry of Health and Family Welfare provided ethical approval for the surveys. The National Institute of Population Research and Training conducted the surveys with the support of the United States Agency for International Development (USAID). The survey protocol was approved by the National Research Ethics Committee in Bangladesh and ORC Macro (Macro International Inc.) Institutional Review Board. Informed consent was obtained from all participants.

## Results

### Trends and projections of U5M

Trends of NM, IM, and U5M in Bangladesh over the survey years 1994–2014 are presented in Table 2. The rates of decline covering these years were around 42% for NM, 48% for IM and 50% for U5M; the annual rate of reduction (ARR) was 2.7% for NM, 3.2% for IM, and 3.4% for U5M. The rates would be 11.4 for NM, 13.7 for IM, and 17.6 for U5M per 1000 live births by 2030 if the current trends continued. When compared to the 1994 estimates, these estimates would be around 77% lower for NM, 82% lower for IM, and 79% lower for U5M with a yearly decrease of 3.9%, 4.6%, and 4.2%, respectively. The estimated U5M rates were always higher in rural areas and among poor households. The projected rates of U5M per 1000 live births by 2030 are 16.9 for urban people, and 21.2 for rural people if the observed trends continued. The U5M rates among women of middle and rich socio-economic status would take the figures to 13.4 and 5.4 deaths per 1000 live births by the year 2030 and such women would experience reductions of 84.0% and 90.8% in U5M by 2030 compared to their experiences in 1994 (Table 2).

### Spatial variations

The rates of NM, IM, and U5M across seven administrative divisions of Bangladesh are presented in Table 3. The rates were calculated using the 2014 BDHS data and were found to be higher for the Sylhet division at around 41.4 for NM; 60.5 for IM and 63.1 for U5M per 1000 live births with lower rates reported for the Barisal division (Table 3).

### Maternal high-risk fertility behaviors and healthcare service use

The trend of maternal high-risk fertility behaviours and use of maternal healthcare services over the period 1994–2014 are presented in Figs 1 and 2. Almost all maternal high-risk fertility

**Table 2. Trends in and projection of under-5 mortality rate (per 1000 live births) according to residence and economic status in Bangladesh over the period 1994–2030.**

| Measures | Observed rate | | | | | | | Change (%) during 1994–2014 | | Projected rate | | Projected change (%) during 1994–2030 | |
|---|---|---|---|---|---|---|---|---|---|---|---|---|---|
| | 1994 | 1997 | 2000 | 2004 | 2007 | 2011 | 2014 | Decline | ARR | 2025 | 2030 | Decline | ARR |
| Neonatal mortality | 48.8 | 46.2 | 41.3 | 40.8 | 36.9 | 31.1 | 28.1 | 42.4 | 2.7 | 16.5 | 11.4 | 76.6 | 3.9 |
| Infant mortality | 76.1 | 80.1 | 64.4 | 63.9 | 50.6 | 40.5 | 39.6 | 47.9 | 3.2 | 23.3 | 13.7 | 81.9 | 4.6 |
| Under-5 mortality | 82.5 | 92.5 | 73.4 | 71.9 | 55.9 | 44.8 | 41.0 | 50.3 | 3.4 | 28.2 | 17.6 | 78.7 | 4.2 |
| **Under-5 mortality by Socio-demographic characteristics** | | | | | | | | | | | | | |
| **Residence** | | | | | | | | | | | | | |
| Urban | 57.7 | 74.1 | 72.9 | 77.9 | 43.5 | 43.1 | 35.0 | 39.3 | 2.5 | 21.8 | 16.9 | 70.7 | 3.4 |
| Rural | 85.2 | 94.3 | 73.5 | 70.5 | 59.4 | 45.3 | 43.0 | 49.5 | 3.4 | 30.3 | 21.2 | 75.1 | 3.8 |
| **Economic status[1]** | | | | | | | | | | | | | |
| Poor | - | - | - | 76.6 | 59.2 | 52.4 | 50.3 | 34.3 | 4.1 | 35.1 | 21.1 | 72.5 | 7.7 |
| Middle | - | - | - | 83.9 | 63.1 | 42.3 | 42.4 | 49.5 | 6.6 | 33.3 | 13.4 | 84.0 | 10.8 |
| Rich | - | - | - | 58.7 | 48.1 | 37.1 | 30.1 | 48.7 | 6.4 | 19.8 | 5.4 | 90.8 | 13.8 |

**Note:** All the rates are weighted. ARR, annual rate of reduction.

[1] Followed standard BDHS measure and trend observed for year 2004 to 2014 due to missing data.

behaviours have seen a slight reduction over these years except for birth order >3 that saw a reduction to 15.2% in 2014 from 35.4% in 1994. There were significant increases in the use of maternal healthcare services such as at least 4 ANC visits (5.4% to 31.8%); institutional delivery (3.8% to 37.9%); CS delivery (3.0% to 23.9%) and delivery by a skilled birth attendant (SBA) (13.4% to 52.9%) over the period 1994 to 2014 (S2 Table).

## Distribution of U5M

In the analysis, most of the women were aged 15–24 years (48.6%), came from poor households (41.9%), lived in rural areas (74.6%), and were unemployed (75.4%). Around 29.1% of the births were unwanted and around 7% of babies were born with VLBW (Table 4)

The distribution of U5M across the variables of maternal high-risk fertility behaviours and use of healthcare services are presented in Table 4. The U5M rate was found to be higher among women whose age at birth was <18 years (6.3%) and a birth interval of <24 months (8.3%). The rate of U5M increased substantially once multiple characteristics of high-risk behaviours were considered together. For instance, around 16% of U5M was found among

**Table 3. Spatial variation on different forms of child mortality (per 1000 live births) in Bangladesh using data from BDHS, 2014.**

| Regions | Types of child mortalities (%, 95% CI) | | |
|---|---|---|---|
| | **Neonatal mortality (NM)** | **Infant mortality (IM)** | **Under-5 mortality (U5M)** |
| *Barisal* | 14.5 (7.5–27.7) | 19.0 (10.6–33.8) | 21.8 (11.9–39.6) |
| *Chittagong* | 24.7 (17.4–35.0) | 45.2 (31.7–64.0) | 48.9 (35.1–67.7) |
| *Dhaka* | 20.4 (13.7–30.3) | 28.9 (20.7–40.1) | 29.6 (21.3–40.9) |
| *Khulna* | 32.6 (21.5–49.1) | 38.3 (26.6–54.7) | 43.8 (30.6–62.4) |
| *Rajshahi* | 36.2 (23.8–54.6) | 41.1 (28.0–60.1) | 42.7 (29.0–62.6) |
| *Rangpur* | 23.4 (15.3–35.7) | 30.6 (20.8–44.8) | 33.8 (23.3–48.7) |
| *Sylhet* | 41.4 (27.8–33.4) | 60.5 (42.2–86.0) | 63.1 (44.6–88.7) |

**Note:** The sample were weighted. Calculated for row percentage and percentages may not total 100.0 because of rounding.

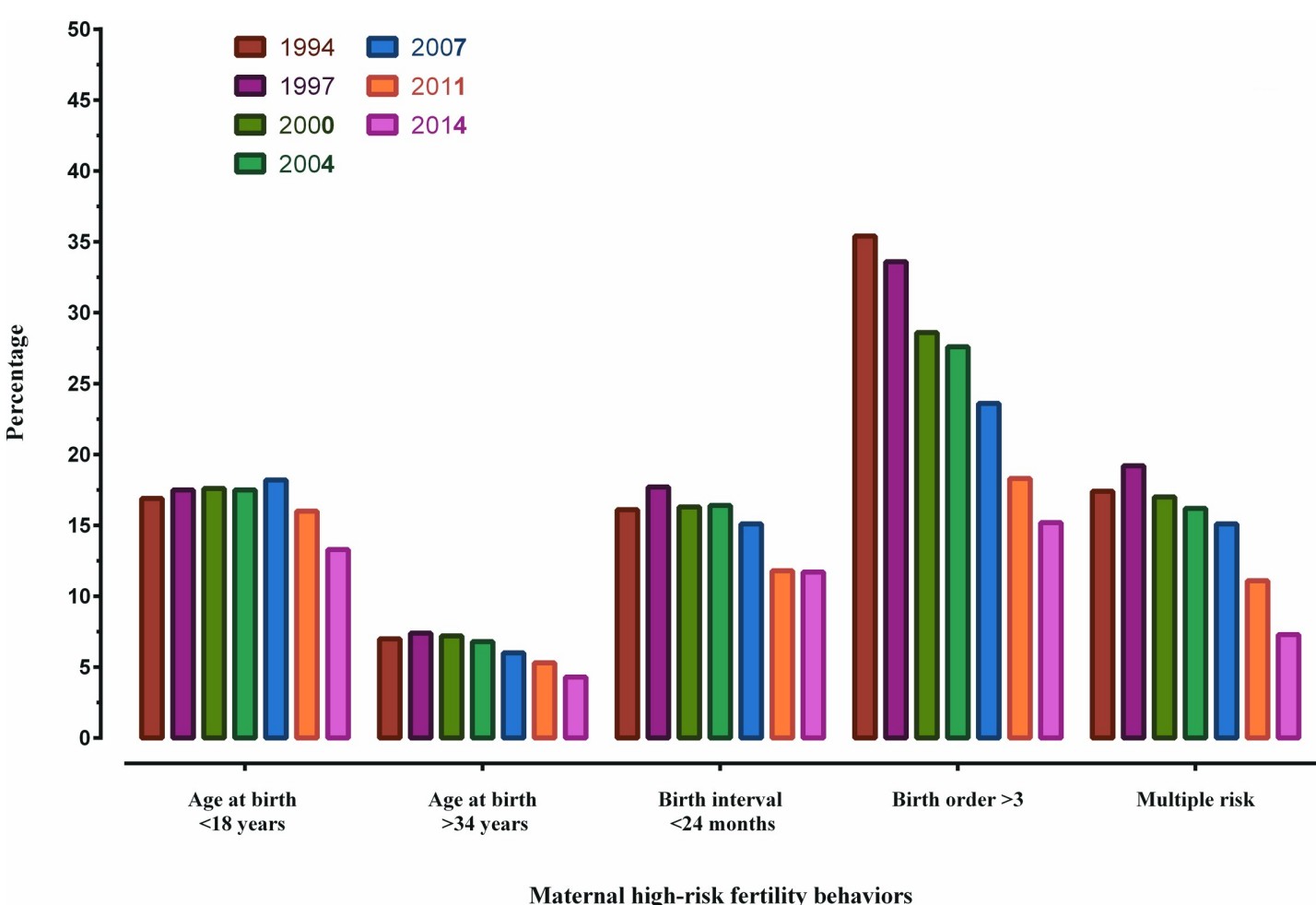

**Fig 1. Trends in maternal high-risk fertility behaviors in Bangladesh over the period 1994–2014.**

women whose age at birth was <18 years and had a birth interval <24 months. Similarly, around 15.2% of U5M was found among women that had a birth order >3 and birth interval of <24 months. The burden was found to be lower among women that had at least 4 ANC visits (2.1%), had SBA services during delivery (2.8%), received CS delivery (2.2%), and had taken up PNC (3.0%). The U5M rate was also found to be higher among unwanted births (5.4%) and children born with VLBW (6.2%) (Table 4).

## Association between selected variables and U5M

The effects of maternal risk-fertility behaviours and use of maternal healthcare services on U5M are presented in Table 5. The risk of U5M was found to be around 1.84 times (95% CI: 1.23–2.76) higher among women aged <18 years at their most recent birth compared to their counterparts. A birth interval of less than 24 months was found to be associated with 1.56 times (95% CI: 1.02–2.37) higher likelihood of U5M than with a birth interval >24 months. The risk of U5M was further increased for those women exhibiting multiple high-risk fertility behaviours. For instance, women aged <18 at their most recent birth and had a previous birth at an interval of <24 months from the most recent birth, was found to have a 226% higher likelihood of U5M (aOR, 2.26, 95% CI: 1.09–4.70) than did their counterparts. The risk was

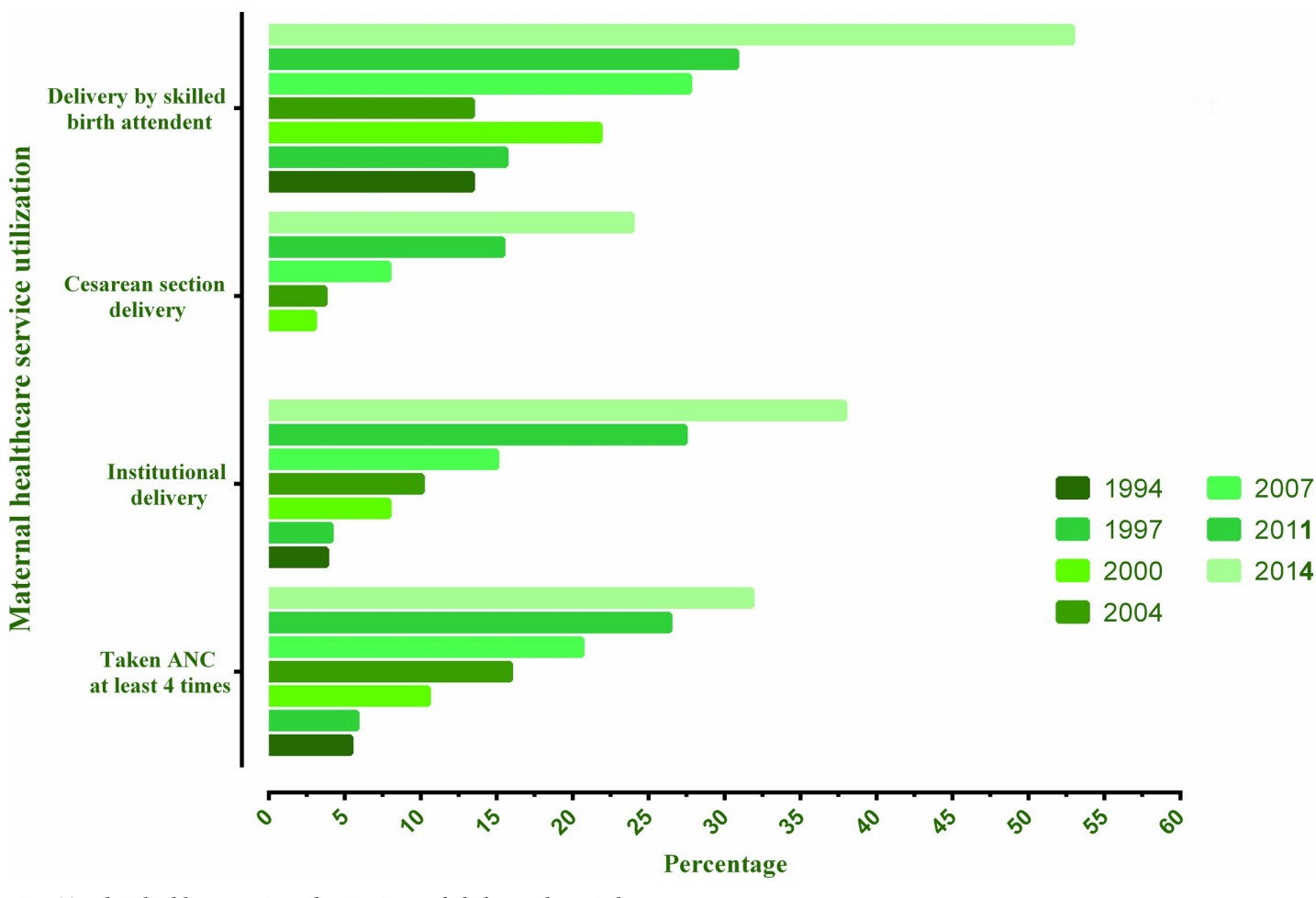

**Fig 2. Trends in healthcare service utilization in Bangladesh over the period 1994–2014.**

around 333% (aOR, 3.33, 95% CI: 1.28–8.70) higher for women that had >3 births with an interval of <24 months in the two most recent successive births, than did their counterparts. This analysis indicates there is a strong positive effect on reducing U5M when healthcare services are used. For example, it was discovered that there is a nearly 50% lower likelihood of U5M among women who had received at least 4 ANC visits and had their delivery by CS (aOR: 0.49; 95% CI: 0.29–0.82) in their most recent birth compared to those that had not. There is evidence that among women that received SBA there was a 53% decline (aOR, 0.47; 95% CI: 0.24–0.95) in U5M cases and a 40% decline in such cases (aOR: 0.61, 95% CI: 0.41–0.91) among women that had received PNC compared to women that had not used these services. The same results were found for NM and IF although these results are not shown (Table 5).

## Discussion

This study has examined the trends and projections of the rate of U5M in Bangladesh and explored the effects of risky maternal fertility behaviours and use of healthcare services on this rate. The U5M rate was found to be gradually declining in Bangladesh and if this decline continues, then it would take the rate to 17.6 per 1000 live births by 2030 that is under the current development goal (SDG) rate of 25 U5M per 1000 live births to be achieved by that year. This

**Table 4. Bivariate distribution of under-5 mortality according to socio-demographic characteristics, maternal risk factors i.e. high-risk fertility behaviors and health-care utilizations based on BDHS-2014.**

| Background characteristics | Frequency in % (95% CI) | Under-5 mortality % (95% CI) | p-values |
|---|---|---|---|
| **Maternal current age (in years)** | | | |
| 15–24 | 48.6 (46.9–50.3) | 4.1 (2.9–5.8) | 0.042 |
| 25–34 | 43.7 (41.9–45.4) | 3.8 (2.3–6.3) | |
| 35–49 | 7.7 (6.9–8.5) | 5.7 (6.4–9.4) | |
| **Maternal education** | | | |
| No education | 16.4 (14.9–18.0) | 4.3 (2.9–6.5) | 0.014 |
| Primary | 27.9 (26.6–29.4) | 4.4 (2.9–6.7) | |
| Secondary and above | 55.6 (53.6–57.6) | 3.9 (2.6–5.9) | |
| **Residence** | | | |
| Urban | 25.4 (21.7–29.5) | 3.5 (1.8–6.7) | 0.152 |
| Rural | 74.6 (70.5–78.3) | 4.3 (2.9–6.3) | |
| **Economic status** | | | |
| Poor | 41.9 (39.4–44.4) | 5.0 (3.5–7.3) | 0.001 |
| Middle | 19.4 (17.9–20.9) | 4.2 (2.3–7.7) | |
| Rich | 38.8 (35.9–41.8) | 3.0 (1.8–5.1) | |
| **Employment status** | | | |
| Unemployed | 75.4 (73.7–77.0) | 4.4 (3.1–6.3) | <0.001 |
| Employed | 24.6 (23.0–26.2) | 3.1 (1.7–5.7) | |
| **Unwanted birth** | | | |
| No | 70.9 (68.6–73.1) | 3.4 (2.2–5.3) | <0.001 |
| Yes | 29.1 (26.9–31.4) | 5.4 (3.6–7.9) | |
| **Very low birth weight (VLBW) of child** | | | |
| No | 93.2 (92.2–94.1) | 3.9 (2.6–5.7) | <0.001 |
| Yes | 6.8 (5.9–7.8) | 6.2 (4.2–9.1) | |
| **Maternal high-risk fertility behavior** | | | |
| **Maternal age at birth <18 years** | | | |
| No | 86.7 (85.8–87.6) | 3.8 (2.5–5.5) | <0.001 |
| Yes | 13.3 (12.4–14.2) | 6.3 (4.3–9.0) | |
| **Maternal age at birth >34 years** | | | |
| No | 95.7 (95.1–96.2) | 4.1 (2.8–5.9) | |
| Yes | 4.3 (3.8–4.9) | 4.9 (2.4–9.8) | 0.745 |
| **Birth interval <24 months** | | | |
| No | 88.7 (87.5–89.8) | 3.4 (2.4–5.1) | |
| Yes | 11.3 (10.2–12.5) | 8.3 (5.5–12.1) | <0.001 |
| **Birth order >3** | | | |
| No | 84.8 (83.5–85.9) | 4.0 (2.7–5.9) | |
| Yes | 15.2 (14.0–16.5) | 4.5 (2.9–7.0) | 0.085 |
| **Maternal age at birth <18 years and Birth interval <24 months[2]** | | | |
| No | 98.9 (98.5–99.2) | 3.8 (2.6–5.5) | |
| Yes | 1.1 (0.7–1.5) | 16.3 (10.3–24.7) | <0.001 |
| **Maternal age at birth <34 years and Birth interval <24 months[3]** | | | |
| No | 95.1 (94.2–95.8) | 4.0 (2.7–5.9) | |
| Yes | 4.9 (4.2–5.8) | 4.6 (1.9–10.7) | 0.590 |
| **Birth interval <24 months and Birth order >3** | | | |
| No | 96.9 (96.3–97.6) | 3.8 (2.6–5.6) | |

*(Continued)*

**Table 4.** (Continued)

| Background characteristics | Frequency in % (95% CI) | Under-5 mortality | p-values |
|---|---|---|---|
| | | % (95% CI) | |
| Yes | 3.02 (2.4–3.7) | 15.2 (8.7–25.1) | <0.001 |
| **Maternal health-care utilization** | | | |
| **Taken ANC at least 4 times** | | | |
| No | 68.7 (66.2–71.1) | 4.9 (3.5–6.7) | <0.001 |
| Yes | 31.8 (28.8–33.7) | 2.1 (1.0–4.5) | |
| **Institutional delivery** | | | |
| No | 62.2 (59.4–64.8) | 4.5 (3.1–6.7) | 0.151 |
| Yes | 37.9 (35.2–40.6) | 3.2 (2.1–4.8) | |
| **Skilled birth attendant (SBA)** | | | |
| No | 47.1 (44.5–49.8) | 5.4 (3.5–8.2) | <0.001 |
| Yes | 52.9 (50.2–55.5) | 2.8 (1.9–4.0) | |
| **Cesarean section (CS) delivery** | | | |
| No | 76.0 (73.8–78.1) | 4.6 (3.1–6.8) | 0.001 |
| Yes | 23.9 (21.8–26.1) | 2.2 (1.3–3.5) | |
| **Post-natal care (PNC)** | | | |
| No | 36.5 (33.5–39.6) | 5.8 (3.8–8.6) | <0.001 |
| Yes | 63.5 (60.4–66.5) | 3.0 (1.9–4.7) | |

**Note:** The sample are weighted. Frequency is calculated for column percentage and percentages may not total 100.0 because of rounding. Percentages of U5M is calculated for row percentage. "No" values for U5M is omitted from table.

[2] Includes the category age at birth <18 years with birth order >3 and age at birth <18 years with interval <24 months and birth order >3.

[3] Includes the category age at birth <34 years with interval <24 months and age at birth <34 years with interval <24 months and birth order >3.

rate is also around five times lower than it was in 1994. The NM and IF are also predicted to be at similar rates by 2030. However, these rates of decline are not happening in all regions of Bangladesh and this study has uncovered evidence of high regional variations in U5M with a higher rate reported in the Sylhet region and a lower rate in the Barisal region. The likelihood of U5M was found to be higher among women that reported one or multiple high-risk fertility behaviour and had not used healthcare services. These findings suggest therefore, that there is a need for region-specific policies where priority is given to women that are engaged in such risky behaviour.

This study has provided evidence of the gradual decline of NM, IM, and U5M rates in Bangladesh over the past 20 years (1994–2014) that is consistent with the global trend [7, 9]. The ARR for U5M was 3.4% that is consistent with a 3.9% ARR at the global level during this period [41]. A global comparable decline of U5M in Bangladesh is good news to report but the higher burden of maternal nutrition, poverty, and urban-rural inequality in use of healthcare services cannot also be reported [25, 42]. This study also identifies evidence of a higher rate of U5M among rural and poor women that has been shown in other studies conducted in similar settings [10, 13, 18, 29]. The current trend for a decline in the U5M rate would enable Bangladesh to achieve the relevant SDG target (SGD 3.2) by 2030, but it would not be experienced across all regions. Therefore, policies are urgently needed to reduce existing inequalities in the use of healthcare services between urban and rural areas and making such services available to all women should be a priority in order to achieve a significant reduction in U5M. These steps would help to speed up the process for achieving the relevant SDG target.

This study also finds there is enormous variation in the child mortality rate across different regions in Bangladesh. For example, NM was found to be higher in the Khulna division, whilst

**Table 5. Association between under-5 mortality and maternal risk factors i.e., high-risk fertility behaviors and health-care utilizations using data BDHS-2014.**

| Measures | Under-5 mortality | |
|---|---|---|
| | **Unadjusted** | **Adjusted** |
| | **OR (95% CI)** | **OR (95% CI)** |
| **Maternal age at birth <18 years** | | |
| No [RC] | 1.00 | 1.00 |
| Yes | 1.72 (1.27–2.31) *** | 1.84 (1.23–2.76) ** |
| **Maternal age at birth >34 years** | | |
| No [RC] | 1.00 | 1.00 |
| Yes | 1.21 (0.57–2.58) | 0.76 (0.15–3.80) |
| **Birth interval <24 months** | | |
| No [RC] | 1.00 | 1.00 |
| Yes | 2.49 (1.90–3.27) *** | 1.56 (1.02–2.37) * |
| **Birth order >3** | | |
| No [RC] | 1.00 | 1.00 |
| Yes | 1.13 (0.71–1.81) | 0.67 (0.32–1.46) |
| **Maternal age at birth <18 years and birth interval <24 months** | | |
| No [RC] | 1.00 | 1.00 |
| Yes | 4.91 (3.00–8.04) *** | 2.26 (1.09–4.70) * |
| **Maternal age at birth <34 years and birth interval <24 months** | | |
| No [RC] | 1.00 | 1.00 |
| Yes | 1.16 (0.46–2.92) | 0.89 (0.15–5.53) |
| **Birth interval <24 months and birth order >3** | | |
| No [RC] | 1.00 | 1.00 |
| Yes | 4.48 (2.51–8.01) *** | 3.33 (1.28–8.70)** |
| **Maternal health-care utilization** | | |
| **Taken ANC at least 4 times** | | |
| No [RC] | 1.00 | 1.00 |
| Yes | 0.42 (0.23–0.78) ** | 0.51 (0.27–0.97) * |
| **Institutional delivery** | | |
| No [RC] | 1.00 | 1.00 |
| Yes | 0.69 (0.49–0.96) * | 1.74 (0.98–3.07) |
| **Skilled birth attendant (SBA)** | | |
| No [RC] | 1.00 | 1.00 |
| Yes | 0.51 (0.37–0.71) *** | 0.47 (0.24–0.95)* |
| **Cesarean section (CS) delivery** | | |
| No [RC] | 1.00 | 1.00 |
| Yes | 0.46 (0.28–0.77) ** | 0.49 (0.29–0.82) ** |
| **Post-natal care (PNC)** | | |
| No [RC] | 1.00 | 1.00 |
| Yes | 0.51 (0.34–0.75) *** | 0.61 (0.41–0.91) ** |

**Note:** High-risk fertility behavior variables categorization followed BDHS standard measure. Model adjusted for regions, residence, education, economic status, unwanted birth, very low birth weight, and all the predictors included in this table (weighted sample). Values with superscript asterisks *,**and *** indicate $p < 0.05$, $p < 0.01$, and $p < 0.001$, respectively. OR: odds ratio; CI: confidence interval; ANC: antenatal care.

both IM and U5M were found to be higher in the Sylhet division. A recent study conducted by Gruebner et al found similar results [43] and another study reported that around 12.8% of areas in Bangladesh failed to meet the MDG target 4 of reducing child mortality although the country taken as a whole did achieve it [44]. Taking these regional variation factors into account, country-level policies and programmes that are focused on the regions should provide effective ways for reducing the rate of U5M and enable Bangladesh to actually meet the relevant SDG targets in all its areas.

This study identifies that maternal high-risk fertility behaviours including childbirth before the age of 18 or after 34 plus a higher birth order and shorter birth interval are very influential in prompting U5M in Bangladesh. Women in the younger age cluster (<18 years) have comparatively immature reproductive systems and are less able to handle any complications during childbirth [45–47], that serves to increase the occurrence of adverse outcomes in the health of newborns [46–48]. Women of a higher age during pregnancy are usually faced with undernutrition as well as with other chronic diseases like diabetes and hypertension. [48] and can generally take less care of themselves during pregnancy [48]. In addition, pregnancies at a later age are mostly unwanted. These behaviours represent independent risk factors for LBW, premature birth, and child malnutrition [46] that further increases the likelihood of U5M [45, 46]. In addition, a shorter birth interval (<24 months) can adversely affect maternal health and nutritional status. These factors can lead to complications during pregnancy as well as child deaths, particularly in the neonatal and infancy periods [19, 49]. Importantly, these adverse consequences are responsible for U5M among women that had more than one high risk fertility behaviour at a time [17, 25, 50].

The use of maternal healthcare services including antenatal, delivery, and postnatal care were found to be important factors for reducing U5M as reported in other studies in LMICs [20, 38, 51]. Using healthcare services during pregnancy ensures that necessary care is provided for mothers and children and help to reduce complications during pregnancy. Women that are educated are knowledgeable about other care and services including vaccination, iron-folic supplements, and tetanus toxoids injections [38, 52]. These further services could help to ensure healthy pregnancies by identifying susceptible risks that could further contribute to a reduction in the occurrence of U5M [20, 26, 38, 52, 53]. The use of healthcare services enables women and service providers to identify risks and the need for CS that could further help to reduce pregnancy complications as well as U5M [54].

This study has a number of strengths and limitations. Firstly, nationally representative data was used along with appropriate statistical methods that took into account complex survey design and sample weights. The trends and projection of U5M were identified by using available survey data that is the first of its kind in Bangladesh. Determining the effects of maternal high-risk fertility behaviours and the use of maternal healthcare services on the rate of U5M, with adjustments to take into account a range of confounders, is also the first of its kind to be carried out in Bangladesh. When taken together, these approaches provide the study with strong foundations for its conclusions. However, this study is also based on secondary data that was collected on the last birth that occurred within the five years preceding the survey dates and with this type of data collection, recall bias is common. Putting forward high-risk fertility behaviours and use of healthcare services as risk factors of U5M gives the study strength, but other factors such as preterm birth, malnutrition and vaccination may also lead to U5M and these need to be given further consideration. Data from the 2014 BDHS was used to assess determinants of U5M that could be considered as dated but this was the most recently available dataset in Bangladesh and the projection of U5M for 2020 was similar to the projection for 2014. This provides validity for the findings of this study.

## Conclusion

The rate of U5M in Bangladesh is decreasing significantly. If this trend continues, the SDG goal concerning U5M is predicted to be achieved by 2030. The trend also postulates existing inequalities in rural-urban differentials and also in economic conditions. However, maternal high risk fertility behaviours and poor use of healthcare services are important predictors of U5M and can impede progress towards SDG 3.2. These findings suggest a need for region-specific policies where priority should be given to women most likely to exhibit high-risk fertility behaviours and not make use of maternal healthcare services.

## Supporting information

**S1 Table. Information on model fittings of linear regression model.**
(DOCX)

**S2 Table. Information about plotting of Figs 1 and 2.**
(DOCX)

## Acknowledgments

The authors are thankful to MEASURE DHS for granting data use permission for this study and also acknowledge the support of the Department of Population Science and Human Resource Development, University of Rajshahi, Bangladesh, where this study was conducted. Authors would like to express their gratitude to the Oxford Institute of Population Ageing, University of Oxford for academic support where Professor Hafiz T.A. Khan is also an Associate Professorial Fellow and also express their gratitude to Dr. Helen Findlay for final checking and editing of the manuscript.

## Author Contributions

**Conceptualization:** Mostaured Ali Khan.

**Data curation:** Mostaured Ali Khan, Golam Mustagir.

**Formal analysis:** Mostaured Ali Khan, Nuruzzaman Khan, Obaidur Rahman, Golam Mustagir, Rafiqul Islam.

**Methodology:** Mostaured Ali Khan, Nuruzzaman Khan.

**Supervision:** Rafiqul Islam, Hafiz T. A. Khan.

**Writing – original draft:** Mostaured Ali Khan, Golam Mustagir.

**Writing – review & editing:** Nuruzzaman Khan, Obaidur Rahman, Kamal Hossain, Rafiqul Islam, Hafiz T. A. Khan.

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
