## [Decision Letter · Decision Letter 0]

4 Nov 2020

PONE-D-20-29506

Trends in, and projections of under-5 mortality (U5M) in Bangladesh: effects of maternal high-risk fertility behaviors and healthcare service utilization on U5M.

PLOS ONE

Dear Dr. Islam,

Thank you for submitting your manuscript to PLOS ONE. After careful consideration, we feel that it has merit but does not fully meet PLOS ONE’s publication criteria as it currently stands. Therefore, we invite you to submit a revised version of the manuscript that addresses the points raised during the review process.

We look forward to receiving your revised manuscript.

Kind regards,

Russell Kabir, PhD

Academic Editor

PLOS ONE

Journal Requirements:

3. Please consider modifying your title to ensure that it is specific, descriptive, concise, and comprehensible to readers outside the field (for example by not using acronyms in the title).

5. Please amend your list of authors on the manuscript to ensure that each author is linked to an affiliation. Authors’ affiliations should reflect the institution where the work was done (if authors moved subsequently, you can also list the new affiliation stating “current affiliation:….” as necessary).

6. Please upload a copy of Figure 2, to which you refer in your text on page 26. If the figure is no longer to be included as part of the submission please remove all reference to it within the text.

8. Thank you for stating the following in the Competing Interests section:

We note that one or more of the authors are employed by a commercial company: MEL and Research.

8.1. Please provide an amended Funding Statement declaring this commercial affiliation, as well as a statement regarding the Role of Funders in your study. If the funding organization did not play a role in the study design, data collection and analysis, decision to publish, or preparation of the manuscript and only provided financial support in the form of authors' salaries and/or research materials, please review your statements relating to the author contributions, and ensure you have specifically and accurately indicated the role(s) that these authors had in your study. You can update author roles in the Author Contributions section of the online submission form.

8.2. Please also provide an updated Competing Interests Statement declaring this commercial affiliation along with any other relevant declarations relating to employment, consultancy, patents, products in development, or marketed products, etc.  

Reviewers' comments:

Reviewer's Responses to Questions

**Comments to the Author**

1. Is the manuscript technically sound, and do the data support the conclusions?

Reviewer #1: Yes

Reviewer #2: Yes

2. Has the statistical analysis been performed appropriately and rigorously? 

Reviewer #1: Yes

Reviewer #2: Yes

3. Have the authors made all data underlying the findings in their manuscript fully available?

Reviewer #1: Yes

Reviewer #2: Yes

4. Is the manuscript presented in an intelligible fashion and written in standard English?

Reviewer #1: No

Reviewer #2: Yes

5. Review Comments to the Author

Reviewer #1: While the statistical analysis is comprehensive, the manuscript requires extensive language editing- which affects the overall readability of the manuscript.

As an example: line 78 '....South Asian countries that placing it in the 3rd 78 positioned country after Pakistan' should probably read

'....South Asian countries which places it in the third position after Pakistan'. There are several other instances of incomplete and and ungrammatical sentences, for example line 79, 90, 92, 140-142 all in the introduction require language editing.

Other than that, the manuscript is sound.

Reviewer #2: In the present work entitled “Trends in, and projections of under-5 mortality (U5M) in Bangladesh: effects of maternal high-risk fertility behaviors and healthcare service utilization on U5M”, authors presented important aspects with sound analytical explanations for the policymakers. Here, the authors projected the under-5 mortality (U5M) in Bangladesh and explored its associations with maternal high-risk fertility behaviors and utilization of available healthcare services. The authors presented a very sound analytical piece of work, and then justified its significance using the real-life application.

I have a major concern, which needs to be addressed in the manuscript.

In line no. 152, authors used the information obtained from the women that “Did you give birth to any children within five years”. In the situations of twins and multiple births within 5 years, how the additional information “Is the child alive?” addresses the living status of children.

Authors should elaborate their explanations of outcome variables for a more practical situation and also how the model incorporated those aspects.

6. PLOS authors have the option to publish the peer review history of their article (what does this mean?). If published, this will include your full peer review and any attached files.

Reviewer #1: **Yes: **Annah Bengesai

Reviewer #2: No

---

## [Author Response · Author response to Decision Letter 0]

25 Dec 2020

Journal Requirements:

Response: We have revised the manuscript as per PLOS ONE’s guidelines.

2.We suggest you thoroughly copyedit your manuscript for language usage, spelling, and grammar. If you do not know anyone who can help you do this, you may wish to consider employing a professional scientific editing service. 

Response: As per your advice one of our colleague, who is native in English (Dr Helen Findlay, not listed in the author’ list) edited the manuscript to correct grammar, and improve readability, clarity and quality of the manuscript.

3. Please consider modifying your title to ensure that it is specific, descriptive, concise, and comprehensible to readers outside the field (for example by not using acronyms in the title). 

Response: We have modified the title. 

 Response: The data underlying this study (BDHSs, 1993-2014) are third party data, and are available from the MEASURE DHS Archive via the instructions included in the following link: http://dhsprogram.com/data/Using-Datasets-for-Analysis.cfm. 

5. Please amend your list of authors on the manuscript to ensure that each author is linked to an affiliation. Authors’ affiliations should reflect the institution where the work was done (if authors moved subsequently, you can also list the new affiliation stating “current affiliation:….” as necessary).

 Response: The affiliations are mentioned accordingly.

6. Please upload a copy of Figure 2, to which you refer in your text on page 26. If the figure is no longer to be included as part of the submission please remove all reference to it within the text.

 Response: Uploaded the figure accordingly.

 Response: Included accordingly.

8. Thank you for stating the following in the Competing Interests section:

We note that one or more of the authors are employed by a commercial company: MEL and Research.

Response: None of the authors have any competing interest in current study. The affiliation of MEL and Research, Practical Action do not have any role in this study.

8.1. Please provide an amended Funding Statement declaring this commercial affiliation, as well as a statement regarding the Role of Funders in your study. If the funding organization did not play a role in the study design, data collection and analysis, decision to publish, or preparation of the manuscript and only provided financial support in the form of authors' salaries and/or research materials, please review your statements relating to the author contributions, and ensure you have specifically and accurately indicated the role(s) that these authors had in your study. You can update author roles in the Author Contributions section of the online submission form.

Response: There was no funding source for this study or the affiliation of MEL and Research, Practical Action do not have any role in this study.

 Response: There was no funding source for this study or the affiliation of MEL and Research, Practical Action do not have any role in this study.

8.2. Please also provide an updated Competing Interests Statement declaring this commercial affiliation along with any other relevant declarations relating to employment, consultancy, patents, products in development, or marketed products, etc. 

Response: The authors received no specific funding for this work, neither have any competing interest. There was no funding source for this study or the affiliation of MEL and Research, Practical Action do not have any role in this study.

Reviewer #1

While the statistical analysis is comprehensive, the manuscript requires extensive language editing- which affects the overall readability of the manuscript.

As an example: line 78 '.... South Asian countries that placing it in the 3rd 78 positioned country after Pakistan' should probably read'.... South Asian countries which places it in the third position after Pakistan'. There are several other instances of incomplete and ungrammatical sentences, for example line 79, 90, 92, 140-142 all in the introduction require language editing.

Other than that, the manuscript is sound.

Response: Thank you for your appreciations and suggestions. As per your advice one of our colleague, who is native in English (Dr Helen Findlay, not listed in the author’ list) edited the manuscript to correct grammar, and improve readability, clarity and quality of the contents.

Reviewer #2:

In the present work entitled “Trends in, and projections of under-5 mortality (U5M) in Bangladesh: effects of maternal high-risk fertility behaviors and healthcare service utilization on U5M”, authors presented important aspects with sound analytical explanations for the policymakers. Here, the authors projected the under-5 mortality (U5M) in Bangladesh and explored its associations with maternal high-risk fertility behaviors and utilization of available healthcare services. The authors presented a very sound analytical piece of work, and then justified its significance using the real-life application.

Response: Thank you for your appreciations

I have a major concern, which needs to be addressed in the manuscript.

In line no. 152, authors used the information obtained from the women that “Did you give birth to any children within five years”. In the situations of twins and multiple births within 5 years, how the additional information “Is the child alive?” addresses the living status of children.

Authors should elaborate their explanations of outcome variables for a more practical situation and also how the model incorporated those aspects. 

Response: The respondents were asked follow-up questions for every child if they have given birth to multiple children within 5 years prior to the survey date. All the predictors are taken in the model was based on existing global literature.

---

## [Editor Report · Decision Letter 1]

15 Jan 2021

Trends and projections of under-5 mortality in Bangladesh including the effects of maternal high-risk fertility behaviours and use of healthcare services

PONE-D-20-29506R1

Dear Dr. Islam,

We’re pleased to inform you that your manuscript has been judged scientifically suitable for publication and will be formally accepted for publication once it meets all outstanding technical requirements.

Kind regards,

Russell Kabir, PhD

Academic Editor

PLOS ONE
---

## [Editor Report · Acceptance letter]

22 Jan 2021

PONE-D-20-29506R1 

Trends and projections of under-5 mortality in Bangladesh including the *effects of maternal high-risk fertility behaviours and use of healthcare services*

Dear Dr. Islam:

I'm pleased to inform you that your manuscript has been deemed suitable for publication in PLOS ONE. Congratulations! Your manuscript is now with our production department. 

Kind regards, 

on behalf of

Dr. Russell Kabir 

Academic Editor

PLOS ONE